# Deep Learning Approaches for Detecting Freezing of Gait in Parkinson’s Disease Patients through On-Body Acceleration Sensors

**DOI:** 10.3390/s20071895

**Published:** 2020-03-29

**Authors:** Luis Sigcha, Nélson Costa, Ignacio Pavón, Susana Costa, Pedro Arezes, Juan Manuel López, Guillermo De Arcas

**Affiliations:** 1Grupo de Investigación en Instrumentación y Acústica Aplicada (I2A2), ETSI Industriales, Universidad Politécnica de Madrid, Campus Sur UPM, Ctra. Valencia, Km 7., 28031 Madrid, Spain; luisfrancisco.sigcha@upm.es (L.S.); juanmanuel.lopez@upm.es (J.M.L.); g.dearcas@upm.es (G.D.A.); 2ALGORITMI Research Center, School of Engineering, University of Minho, 4800-058 Guimaraes, Portugal; ncosta@dps.uminho.pt (N.C.); susana.costa@dps.uminho.pt (S.C.); parezes@dps.uminho.pt (P.A.)

**Keywords:** IMU, accelerometer, convolutional neural networks, LSTM, consecutive windows, denoising autoencoder, time distributed, spectral representation

## Abstract

Freezing of gait (FOG) is one of the most incapacitating motor symptoms in Parkinson’s disease (PD). The occurrence of FOG reduces the patients’ quality of live and leads to falls. FOG assessment has usually been made through questionnaires, however, this method can be subjective and could not provide an accurate representation of the severity of this symptom. The use of sensor-based systems can provide accurate and objective information to track the symptoms’ evolution to optimize PD management and treatments. Several authors have proposed specific methods based on wearables and the analysis of inertial signals to detect FOG in laboratory conditions, however, its performance is usually lower when being used at patients’ homes. This study presents a new approach based on a recurrent neural network (RNN) and a single waist-worn triaxial accelerometer to enhance the FOG detection performance to be used in real home-environments. Also, several machine and deep learning approaches for FOG detection are evaluated using a leave-one-subject-out (LOSO) cross-validation. Results show that modeling spectral information of adjacent windows through an RNN can bring a significant improvement in the performance of FOG detection without increasing the length of the analysis window (required to using it as a cue-system).

## 1. Introduction

Parkinson disease (PD) is a neurodegenerative illness that causes progressive deterioration of the motor system caused by a loss of dopamine neurons from the substantia nigra of the midbrain [1,2,3,4]. PD is the second most common neurodegenerative disease, and it is characterized by the presence of motor symptoms as resting tremors, stiffness of the trunk and limbs, slowness of movement (bradykinesia), paralysis (akinesia), and postural disorders [5,6].

On a global scale, it is estimated that there are 7 to 10 million people living with this disease. Its prevalence increases with age, being rare before the age of 50 [7,8].

Other common motor manifestations in PD are variations in stride length and gait speed, dyskinesia, freezing of gait (FOG) and falls associated with motor problems. Gait difficulties and postural instability make PD highly incapacitating, making it difficult to sit and stand up and forcing a gait with small steps without the normal pendulum movement of the arms [2,3].

FOG is one of the most disabling symptoms in PD. The symptoms are usually observed in the advanced stage of the disease, approximately 50% of all PD patients have some symptoms and about 80% are severely affected [9,10,11,12].

FOG events typically manifest as a sudden and transient inability to move, typically occuring on initiating gait, on turning while walking, or when experiencing stressful situations. During FOG episodes PD patients report that their feet are inexplicably glued to the ground [13].

When a FOG event appears in walking, patients show variability in gait and a considerable reduction in step length and frequently present trembling of the legs [11,12]. While normal gait steps have a main frequency from 0.5 to 3 Hz (determined with sensors placed on the ankles), FOG events present a main frequency in the range from 6 to 8 Hz [14,15,16]. 

Because of the unexpected nature, FOG events are difficult to assess and often produce falls with a significant reduction in the quality of life [17,18,19,20]. The current FOG treatment to alleviate the symptoms includes physical rehabilitation, brain stimulation, and medication [21].

The methods employed to evaluate the presence of FOG are based on specific gait tests or using questionnaires about the frequency and severity of FOG episodes [22,23,24]. 

This kind of assessment provides relevant indicators for the identification and characterization of FOG. However, these methods present drawbacks because they are performed in a clinical setting, which does not reflect real-life settings, such as the patients’ homes, where FOG events tend to occur more frequently and may present different patterns, especially when patients are performing activities of daily living (ADL) [6,13,25]. 

Wearable devices and portable sensors are discreet and comfortable options to collect and measure physical parameters such as movement, body temperature, and heart rate, among others. In recent years, there has been a large growth in this technological niche, providing wearables functionalities currently used in many applications such as health, wellness, safety, sports, fitness, communication, and business [26,27]. Machine learning (ML) techniques have been used in conjunction with wearables for activity detection and the combination of these technologies has led to an unprecedented performance in a variety of applications [28,29,30,31,32,33,34].

To improve the ability to detect and control the occurrence of fluctuations in the motor system in PD, new technological developments have been proposed to improve the management of PD through wearable devices and ML. These kinds of technologies can assist medical practitioners to make objective decisions regarding medication type and dose adjustments, allowing the development of more effective treatments that increase the patient’s quality of life and reduce costs related to medical care [35,36,37,38,39].

The assessments through wearable devices can provide accurate, unobtrusive, and long-term health data to ensure optimal therapy. Automatic FOG analysis and detection systems could provide to practitioners with relevant information about the occurrence and evolution of the FOG events [40].

An accurate system could be used for reducing the duration and frequency of FOG episodes by using rhythmic auditory cues which have shown an improvement in walking by maintaining the speed and amplitude of movements [41,42,43] while a low latency FOG detection system could increase effectivity when applied only during FOG episodes [43,44]. 

Recent studies have started to use deep learning (DL) for automatic classification. DL is a field of artificial intelligence that employs techniques capable of extract automatically discriminant characteristics from data (e.g., from signals obtained directly from sensors without any pre-processing). DL has enabled the development of end-to-end classifiers and has shown unprecedented performance in many applications such as image recognition, computer vision, medical information processing, bioinformatics, natural language processing, logical reasoning, robotics, and control, etc., [29,30,31,32,33,34,45]. For this reason, DL methods have been adopted to work in systems designed for human activity recognition (HAR), using information obtained from multiple sensors [30,31,32,33,34].

DL is a representation-learning method based on complex networks composed of several processing layers to learn representations of data with multiple levels of abstraction [29]. By stacking several intermediate layers, it is possible to create highly abstract functions that automatically extract high-level features in data to identify activities such as walking or running [33,45,46].

DL is characterized by a wide variety of techniques and architectures. From the available literature, convolutional neural networks (CNN) have been identified as excellent candidates to be combined with other discriminative networks such as full connected neural networks [31,33,47].

CNN is a kind of deep neural network, that has the capability to work as a function extractor. These types of networks can be applied in temporary signals and images to work as a type of filter that automatically extracts discriminant characteristics. This type of network can group several convolutional operators to create some hierarchical and progressively more abstract features that are learned automatically [48].

Recurrent neural networks (RNN) are a type of neural network that exhibits temporary behavior because the connections between nodes simulate a discrete-time dynamical system [49]. Even though the convolutional neural network architecture is excellent for extracting invariant local characteristics, it becomes ineffective to model data with global time dependencies. However, the RNN is designed to work with time series data such as those obtained from sensors [50].

At present, long short-term memory (LSTM) is the most commonly used type of RNN. These networks are characterized by including a memory space controlled through different gates, which allows modeling time dependencies in sequential data [51].

The combination of CNN and LSTM layers in the same deep network has established the state-of-the-art in areas like speech analysis, natural language processing, and motion recognition, where it is necessary to model temporal information. By using this type of combined architecture, it is possible to capture temporal dependencies in features extracted from convolutional layers [34,45].

This paper evaluates some classification and detection techniques based on machine and DL with accelerometer signals acquired from a body worn IMU to detect FOG in Parkinson’s patients in home environments. To make a baseline, three data representations proposed in the literature have been reproduced, which includes: Mazilu features [44], Mel frequency cepstral coefficients (MFCCs) adapted to inertial signals [52], and fast Fourier transform (FFT) [53]. These data representations were tested by using a Random forest [54] classifier and random 10-fold cross-validation (R10fold) and leave-one-subject-out (LOSO) cross-validation to compare the results with previous work and to find an accurate data representation to test the DL approaches.

For testing the DL approaches, three architectures were evaluated: a denoiser autoencoder [55], a deep neural network with CNN, and a combination of CNN and LSTM layers. For comparison proposes, shallow algorithms such as one-class support vector machines (OC-SVM) [56], support vector machines (SVM) [57], AdaBoost [58], and random forest were tested.

This study was performed using the data collect by Rodríguez-Martín et al. [59]. This data includes recordings from 21 PD patients who manifested FOG episodes when performing ADL at their homes.

The best result achieved shows an improvement from 0.904 to 0.939 of the area under the curve (AUC) with a LOSO evaluation. The results show that the best method is able to outperform the state-of-the-art on FOG detection by using a single triaxial accelerometer and without adding delay or increasing the size of the temporal analysis window.

## 2. Related Work

In this section, a review of the related work to FOG detection by using body-worn sensors is performed. According to the consulted literature, several devices and algorithms have been proposed to increase the accuracy in the detection of FOG events [14,15,44,59,60,61,62,63,64,65,66,67].

Most authors have used R10fold and LOSO cross-validation to evaluate the performance of their systems with standardized metrics. Whereas other authors have kept a certain number of patients for evaluating their systems. Common evaluation metrics in FOG detection are specificity and sensitivity. The specificity is the ratio of negatives that are correctly identified, while sensitivity is the ratio of positives that are correctly identified.

While in R10fold cross-validation the data of 1 patient is randomly divided into 10-folds and then 9-folds are used for training and 1 for testing. LOSO cross-validation consists on creating a training subset that includes data from all patients except one that will be used for testing [59,64,67].

Table 1 summarizes previous highlighted works regarding FOG detection with wearable devices, the location of the sensors, the number of participants and the main results.

The symptomatology of FOG manifests differently among patients and is strongly dependent on the context. Freezing events can generally occur randomly. Situations in which a patient manifests FOG are different from the conditions in which other patients can present FOG. However, in terms of inertial signals and its spectral information, some common characteristics associated with FOG events can be used for automatic detection [64].

In 2008 Moore et al. [14] proposed a threshold method for automatic FOG detection by using a spectral analysis of specific bands from signals of a microelectromechanical system (MEMS) accelerometer. The authors introduce the freeze index (FI) that is the ratio between the power of the freeze-band (3–8 Hz) and the locomotion band (0.5–3 Hz).

The sensor was placed at the left shank of 11 PD patients who manifested FOG episodes and the analysis was made by splitting the signals into temporal windows of 6 seconds. The reported results of their system were the detection of 78% of the FOG events. However, the data employed contained only 46 FOG events and the authors obviated the rate of wrong FOG predictions. 

In 2009, Bächlin et al. [60] proposed the addition of a second threshold to the Moore et al. [14] method. The new threshold uses the power band in the range of 0.5–8 Hz to avoid false detections when standing. A second improvement was the reduction of the temporal window to 4 seconds in order to reduce latency time.

Bächlin’s method was evaluated on the Daphnet Freezing of Gait Dataset [68] which contains inertial signals of three MEMS triaxial accelerometers placed on the shank, the thigh, and the lower back. The data collection was made under controlled conditions in clinical facilities with the participation of 10 patients, from whom 8 manifested FOG episodes. The features extracted from 4 seconds windows were the power spectra of the locomotion band (0.5–3 Hz), the freezing band (3–8 Hz), and the total power between 0.5–8 Hz.

The best results reported in [60] were 73.1% for sensitivity and 81.6% for specificity in FOG detection.

In 2012, the work of Mazilu et al. [44] presented for the first time a FOG detection system based on machine learning algorithms. The authors tested the use of different temporal windows and the performance of several classification algorithms using the Daphnet dataset. In this study, the features used for classification were based on Bächlin’s work [60] and additional features such as mean, standard deviation variance and entropy extracted from the acceleration signals.

The reported results for a user dependent model (10-fold cross-validation) were a sensitivity and specificity over 95% in most cases using a temporal window of 4 seconds. Meanwhile, the best result reported for a generic model (LOSO cross-validation) was 66.25% for sensitivity and 95.38% for specificity when using Random forests as a classifier with windows of 4 seconds.

Later in 2013, Moore et al. [61] made a study to analyze the use of seven sensors in different positions of the body with Bächlin et al. [60] features and temporal windows with different lengths. The study was carried with inertial signals from 25 PD patients. The best performances were achieved using windows with lengths from 2.5 to 5 seconds using all sensors. Besides, the shank and back were identified as the most convenient places to put a single sensor, showing similar performances to the seven-sensors configuration.

In the same year (2013), a system for automatic FOG detection was proposed by Tripoliti et al. [62]. The system was based on four steps: data imputation (interpolation), band-pass filtering, entropy calculation, and automatic classification.

The data in this study was collected from 16 people using 6 accelerometers and 2 gyroscopes attached to different body parts. The results reported with LOSO evaluation and temporal windows of 1 second with an overlap of 50% were 89.99% and 93.48% for sensitivity and specificity considering all test subjects (5 healthy, 6 PD with no FOG, and 5 PD with FOG). However, considering only data from patients who present FOG events the results decrease to 89.3% and 79.15% for sensitivity and specificity. 

In 2015, Zach et al. [63] conducted a study in FOG detection by using a single accelerometer placed at the waist. The study included 23 patients who manifested FOG episodes when performing walking tasks like rapid full turns or rapid short steps.

The reported results were 75% for sensitivity and 76% for specificity considering all proposed walking tasks. In this work, a high performance was achieved using a single accelerometer located in the waits, as suggested by Moore [61].

In 2016, Ahlrichs et al. [15] proposed a method to monitor FOG episodes in a home environment based on acceleration signals collected from 20 patients with a waist-worn device. Several frequency-related features were extracted using a temporal window of 3.2 seconds.

Their best results with an SVM using 10-fold cross-validation were a geometric mean (GM) between specificity and sensitivity of 0.961 (sensitivity of 0.923, and specificity of 1) using data from five patients for testing. This approach shows that frequency-based features could be used reliably to detect FOG using a single sensor.

Later in 2017, a system for FOG detection during the performance of normal ADL was proposed by Rodríguez-Martín et al. [59]. This system was developed and tested considering its use in real life situations, in contrast to previous works made in laboratory environments.

The data were collected by a single IMU sensor placed at the left side of the waist from 21 PD patients who manifested FOG events while performing ADL at their homes. The authors proposed the use of an SVM classifier and a data representation based on 55 features which include statistical, and spectral features. The best achieved results were 88.09% and 80.09% for sensitivity and specificity for the personalized model (R10fold), and a sensitivity of 79.03% and specificity of 74.67% for the generic model tested through LOSO evaluation.

Conversely, in 2017, Samà et al. [64] presented an improvement in the previous system proposed by Rodríguez-Martín et al. [63]. In this work, the authors reduce systematically the number of features from 55 to 28, in order to reduce the computational burden and optimize the classifier to be used in real-time. Several machine learning algorithms were tested with data from 15 PD patients who manifested FOG.

The best results reported were 91.81% on sensitivity and 87.45% on specificity with 10-fold cross-validation. Using LOSO evaluation, the authors report 84.49% on sensitivity and 85.83% on specificity, with an SVM classifier and windows of 3.2 seconds. The proposed data representation presents advantages over previous handmade feature extraction methodologies and shows opportunities for the improvement of FOG detection systems to be applied in real time.

According to the literature, recent studies propose the use of DL models for HAR and FOG detection. When working with sensor signals authors have successfully used deep networks with CNN and fully connected neural networks, while CNN work as an automatic feature extractor, the fully connected layers are used for classification [31,32,33,34].

Regarding FOG, in 2018, Camps et al. [65] presented a method for FOG detection in home environment based on deep learning techniques with an eight-layered model with 1D convolutional layers. The model proposed was trained using 9-channel signals recorded from a waist-worn IMU with three tri-axial sensors: accelerometer, gyroscope, and magnetometer. The authors proposed the use of novel data representation named “Spectral window stacking” based on the spectral analysis through FFT and considering information from one previous temporal window. The authors also proposed the use of data augmentation technics in the training process, based on shifting and rotating some temporal windows for improving the model performance and generalization. The results presented were evaluated by using a split of data to train, validation, and test (instead of LOSO cross-validation). The best results show a performance of 90.6% for the GM, an AUC of 0.88, a sensitivity of 91.9, and a sensibility of 89.5 when using data of four patients for testing.

In the same year 2018, Mohammadian et al. [66] proposed a novelty detection framework based on convolutional denoising autoencoder, to detect atypical movement patterns in patients with Autism Spectrum Disorder and PD, via wearable sensors and a probabilistic self-supervised deep network.

The authors proposed a method to detect abnormal movement without the need for labeled data for training. According to the framework, a movement disorder is identified as a deviation from the normal movements. A probabilistic convolutional denoising autoencoder is used to learn the distribution of the normal human movements from signals of a tri-axial accelerometer to quantify the deviation of each unseen sample from the normal movement samples by using a normative probability map (NPM) proposed in [69].

For evaluating the proposed framework, the authors used the Daphnet dataset [68]. The results showed a superior performance (average AUC of 0.77) when comparing the probabilistic convolutional denoising autoencoder with other self-supervised methods such as OC-SVM and a reconstruction-error denoiser autoencoder. However, despite the excellent results when comparing with supervised methods, the novelty methods show a slightly lower performance.

In 2019, San-Segundo et al. [67] presented a study to increase the robustness of the systems of FOG detection. In this work, four data representations and several deep and ML methods were analyzed, by using a LOSO cross-validation to show results closer to a real situation, where the system must be able to analyze patients whose data have not been used in the training stage.

This evaluation of the proposed methods was performed using the Daphnet dataset. The data representation methods tested in this study include features proposed by Mazilu et al. [44], time and frequency features used in HAR [67], features adapted from the Speech quality assessment (SQA) metrics [70], and Mel Frequency Cepstral Coefficients (MFCCs) adapted to inertial signals [52]. The classification methods were based in a Random forest, multilayer perceptron, hidden Markov models (HMMs), and deep neural networks with convolutional layers.

The results achieved show a significant improvement in the FOG detection when information of previous windows is included in the analysis. The best results with LOSO evaluation were achieved using the current window and three previous windows, with a data representation composed of Mazilu features [44] and MFCCs adapted for inertial signals [52]. While the best classifier was a deep network based on convolutional and full connected layers, the reported results show an AUC of 0.931 and an EER of 12.5%. The authors also identified that the use of a single triaxial accelerometer located in the knee can perform very closely to a system that uses three accelerometers located in different parts of the body.

## 3. Materials and Methods

This section details the dataset used to test the proposed algorithms, as well as the signal pre-processing techniques and the feature extraction techniques corresponding to each of the four data representations.

The first data representation was a set of hand-crafted features related to FOG [44]. The second data representation was composed of MFCCS adapted to inertial signals [52]. The third is a spectral data representation based fast Fourier transform similar to that used in Camps et al. [65]. Lastly, a fourth spectral data representation based on the stacking of a current spectral window and up to three previous analysis windows are tested. As in [67] from here onwards we will be referring to this as such contextual windows. 

For testing the performance of the classification algorithms, first, we compared: OC-SVM and MPL Denoiser Auto-encoder for novelty detection; and then supervised methods were tested such as Random forest, Ada-boost, SVM, deep neural networks with convolutional layers (CNN), and deep networks with convolutional and LSTM layers (CNN-LSTM). The CNN-LSTM model was also used for the analysis of data representation with previous temporal windows.

### 3.1. FOG Data Collection

The data used for testing the proposed methods in FOG detection were collected for the article [59]. The data are composed of acceleration signals from 21 patients (3 women and 18 men) with FOG symptoms and a mean age of 69 years. The data were collected through a wearable IMU located at the waist, in both ON and OFF states.

All data collection was carried out at the patients’ homes. Two tests were performed on the participants, doing scripted ADL with an approximate duration of 20 minutes each. With this approach of free-living, an increase in the time and frequency of the FOG episodes is expected [25,71].

The data were labeled by an experienced clinician using video recordings of the tests. Criteria for the inclusion of patients were: a score higher than 6 on the FOG Questionnaire (FoG-Q) [23], a score greater than 2 in the OFF state according to the Hoehn and Yahr scale [72].

The results in the FOG-Q were a mean index of 15.2 (standard deviation: 4.47) with a range from 6 to 23. While the mean Hoehn and Yahr scoring in OFF state was 3.07 (standard deviation: 0.426).

The ratio between FOG and normal ADL movement is 10 to 1, where the 89.5% corresponds to the absence of FOG symptoms and the remaining 10.5% correspond to FOG events. According to this ratio, the dataset can be considered as a binary unbalanced classification problem.

### 3.2. Signal Preprocessing

This subsection presents a description of the pre-processing applied to the inertial signals, which consist of resampling, filtering, and windowing.

The dataset comprises three raw accelerometer signals with a sampling rate of 200 Hz. For this study, the raw accelerometer signals have been resampled at 40 Hz since this frequency has been shown to be enough to analyze human movements according to Zhou [73]. Furthermore, the use of 40 Hz of sampling rate is enough to analyze the frequency spectrum (below 20 Hz) during FOG episodes according to Moore [14].

Once resampled, the signals were filtered to remove the noise at high frequencies with a second-order Butterworth low-pass filter with a 15 Hz cut-off. Also, to remove the influence of gravity, a third-order Butterworth high-pass filter with a 0.2 Hz was used.

For windowing the signal, the literature indicates that the best performances in FOG detection have been obtained with the use of longer temporal windows. Whereas, when using a short window size, it has been reported an important decrease in the sensitivity at short episodes, which are normally the most frequent in patients with FOG [61,68]. Mazilu [44] and Zach [63] have reported that using 3-second windows or window between 2 and 4 seconds may enable the achievement of similar performances that to longer windows.

In order to be able to use FOG detection as a cueing system to help the patient during FOG episodes to maintain a certain speed by using auditory, visual, or somatosensory feedback it is necessary to apply the cues only when the FOG event appear. Therefore, for making a usable FOG detection system it is necessary that event detection must be fast without adding delays to the detection process [67,74].

In this study, a temporal window of 128 samples is considered to reduce system latency in FOG detection and prevent a heavy computational burden due to the feature extraction. With a sampling rate of 40 Hz, a window with a length of 3.2 seconds is obtained.

To address the reduced amount of FOG events in the data and to achieve a windowing strategy similar to that reported in the literature [65,67] a window was labeled as a FOG only if more than 50% of its samples were labeled as FOG. Whereas to considered it as a non-FOG window, all samples must be labeled as not FOG. Windows containing FOG samples with less than 50% were discarded.

For the first experiment, the accelerometer signals were windowed with three different overlap settings: 0%, 50%, and 75%. The overlap has been applied to verify the behavior in the performance of the FOG detection and to prevent the loss of information between windows due to the feature extraction method. In the rest of the experiments made in this paper, an overlap of 75% has been used, with 25% of the advance.

### 3.3. Data Representations

In order to test different classification methods based on machine learning and DL techniques, diverse feature extraction techniques proposed in the literature have been tested. The different features of these data representations are the input for the proposed classifiers in the training and testing processes. Hand-made features obtained from the acceleration signals have been selected to make a baseline, also spectral features proposed in the literature were resorted to.

The first set of features used contain seven features proposed by Mazilu et al. [44]. When using three acceleration signals, a 21 (7 × 3) feature vector was obtained for each 128-samples window. The second set of features was composed of MFCCs adapted to inertial signals proposed in [52]. In this approach, 12 coefficients were considered for each of the three acceleration channels. With three acceleration signals, a vector of 36 (12 × 3) features is obtained for each window. The third set of features was composed of the symmetric part (64-bin) of a 128-point fast Fourier transform (FFT) [53]. With three acceleration signals, a 192 (64 × 3) features vector for each temporal window was obtained.

An additional data representation was tested by using the features of the FFT and considering previously analyzed windows (contextual windows). With three acceleration signals, a 384 (2 × 64 × 3) features vector was obtained, considering the current FFT and one previous window. Then, a 576 (3 × 64 × 3) features vector considering the FFT two previous windows was attained. Finally, a 768 (4 × 64 × 3) features vector considering the current FFT and the three previous windows was built. A summary of the data representation tested in this work is shown in Table 2. The process to obtain the data representation based in the FFT spectrum and previous windows is shown in Figure 1. 

### 3.4. Tested Algorithms

For testing the proposed classification methods, two novelty detection methods and five supervised algorithms were analyzed.

For novelty detection, an OC-SVM and a reconstruction-error denoiser autoencoder [75] were tested. These detection methods have the advantage of not requiring labelled data for training.

While for supervised methods, several algorithms were analyzed, which include: Random forest with 100 decision trees, AdaBoost with 100 decision trees, support vector machine classifier, deep neural networks with convolutional layers (CNN), and a deep network composed of convolutional and LSTM layers (CNN-LSTM). The CNN-LSTM model allows analyzing sequential data by using previous windows in order to take advantage of temporal behavior found in FOG events. In the next sub sections, the architectures used for the deep-learning based models are shown.

#### 3.4.1. Architecture for ML Denoiser Autoencoder

For the novelty detection based in DL, a denoiser autoencoder composed of seven fully connected layers, according to the scheme presented in Figure 2, was tested. The encoder is composed of three fully connected layers with 192, 80, and 40 neurons with a hyperbolic tangent (tanh) activation function. Then, the encoder is connected to an intermediate layer of 20 neurons with a rectified linear unit (ReLU) activation. While, the decoder is composed of three fully connected layers with 40, 80, and 192 neurons as output is a layer with a tanh activation function.

The train and test data were normalized from 0 to 1 by using the max value of the training subset. The noise factor used in the training subset was set to 0.2.

The selected loss function was the binary cross entropy and the optimizer was the root-mean-square propagation (RMSprop) [76]. The novelty detection is made by the reconstruction error by using the root mean square error (RMSE).

The best model trained used 250 epochs with a batch size of 512. Further analysis of the methodology used for the adjustment of the training parameters (hyperparameter) is explained in Section 3.5. 

#### 3.4.2. Architecture for the Deep Neural Networks with Convolutional Layers (CNN)

For the model based in fully connected neural networks and convolutional layers, two 1D-CNN layers with max-polling for automatic feature extraction were used, while three fully connected layers were used for classification according to Figure 3. 

The first convolutional layer has 32 filters and a kernel size of 8. The second layer has 32 filters and a kernel of 8, the pool size in the max-pooling layers was set 2 in all cases. In order to connect the convolutional to the fully connected layers, a flatten layer was used in between.

After the convolutional layers, three dense layers with dropout were connected. The first dense layer has 128 neurons with a dropout of 0.5, then a layer of 32 neurons with a dropout of 0.3, and the output layer uses 1 neuron. All layers in the model use ReLU as activation functions, except the last one that uses a Sigmoid activation to output the probability of a FOG event.

The configuration that achieved the best performance in the training process was 150 epochs, batch size equal to 512. The loss function was the binary cross entropy with Adam optimizer [77].

#### 3.4.3. Architecture for the Deep Neural Networks with Convolutional and LSTM Layers (CNN-LSTM)

The CNN-LSTM model works as a classifier using sequential information obtained from previous spectral windows of the FFT. The CNN-LSTM network is composed of three main parts: first, a time-distributed layer that works as a wrapper for convolutional layers with max-polling; then, one LSTM layer for sequential modelling and; third a group of dense layers for classification. The proposed CNN-LSTM model is shown in Figure 4.

For the first part, a time-distributed [78] layer which contains two 1D-convolutional layers with max-polling was used to apply convolution to every temporal slice of the input. The convolutional layers have 128 and 64 filters respectively, with a kernel size of 4 in both layers. The max-polling layers have a pool size of 2 in both layers. In these layers, ReLU was resorted to as the activation function. In order to connect the convolutional layers to the LSTM layer, a time-distributed flatten layer was used.

The second part of the model is composed of an LSTM recurrent layer with 64 cells and a tanh activation function. This layer allows modelling the temporal dynamics in time series problems increasing the discriminative power of the network [34].

The third part of the deep network is composed of three fully connected layers for classification. The sixth layer has 80 neurons, the seventh layer has 40 neurons, both ReLU activation, while the octave layer has one neuron with sigmoid activation.

The configuration that achieved the best performance in the training set with previous windows was 150 epochs and a batch size of 1024. The loss function used was the binary cross-entropy with Adam optimizer [77].

### 3.5. Hyperparameter Adjust and Training

Deep neural networks are hard to train because of a large number of parameters that need to be adjusted in each layer [79]. When training a DL model some parameters such as the number of hidden layers, the number of neurons in each layer, the learning rate or the activation function remain constant during training and testing. An accurate selection of these hyperparameters controls the training process in terms of computational processing and has a powerful influence on the performance of the model.

Training of DL models is an iterative process that repeats until finding an acceptable solution to a problem. When training a model, one passes over the entire dataset is named epoch. The correct adjust of the number of epochs is important to prevent overfitting and unnecessary computing.

Another important hyperparameter for training a model is the batch size, which is the number of samples that are processed independently. After every batch, the algorithm performs weight corrections to the neurons, rather than applying a single update per epoch.

In practice, the use of larger batch sizes has shown a reduction in the ability of the model to generalize solutions and require a higher amount of memory to process the data [80]. Therefore, keeping a reduced batch size can help to achieve better performance on generic models.

For this paper, the Hyperopt library was used for hyperparameter optimization [81]. During this process, the following hyper-parameters were tuned: learning rate, number of LSTM cells, the number of CNN and fully connected layers, the number of filters and size of the kernels in CNN layers, the batch size and the number of the neurons in the fully connected layers.

After fine tuning the hyperparameters, the DL models were trained via backpropagation using the following configuration: the optimization method was set to Adam, the learning rate was set to 1 × 10^−4^, a gradient clipping was implemented with clip value set to 1. 

Also, preliminary tests with training samples were used to adjust the maximum number of epochs. Initially, this value was set to 250 with early-stopping when the loss value did not decrease for 20 continuous epochs. Then, this value was systematically reduced in each model to prevent overfitting in validation and avoid unnecessary computing during training.

The batch size was set to 512 for the denoiser auto encoder and the CNN model, and 1024 for the CNN-LSTM model. The lowest batch size with the best performance in the preliminary test was selected to prevent memory problems in training.

## 4. Experiments

In this section, the results achieved for the different models and data representation are reported.

Several approaches to detect FOG based on ML and DL from the literature were reproduced and evaluated to find an accurate data representation to be used for testing the proposed supervised and self-supervised DL models.

Accurate data representation is required to develop a reliable FOG detection method that can be used as part of a wearable (non-intrusive) system that can provide precise monitoring of symptom evolution in home-environments during patients’ daily lives. Also, a reliable FOG detection system should have a reduced latency time to allow the use of non-pharmacologic solutions like external cues to help to reduce the occurrence of FOG episodes.

In the literature, the use of longer temporal windows has shown an improvement performance in FOG detection, while increasing the latency time. This situation makes these systems ineffective to be used as external cues. These considerations make the FOG detection a trade-off between performance and latency.

The aim is to increase the detection performance while maintaining a reduced window size (to reduce the latency). In Section 4.4, a novel (supervised) approach based on LSTM and CNN layers is evaluated. This DL model is able to analyze the temporal dependencies of a spectral data representation including several previous windows, in order to improve the classification capabilities by taking advantage of the spectral information that appears in initial phases of FOG [82,83].

Most of the experiments in this study were carried out with a LOSO cross-validation. This evaluation method has been proposed by several authors in order to develop and test generic FOG detection systems [59,64,67].

In order to make a baseline and for comparison with previous work, three state-of-the-art data representations were tested, with different overlap settings when using a single Random forest classifier. The performance in the baseline was tested by means of random 10-fold cross-validation and with a LOSO evaluation. Also, a single data representation (FFT) was selected from the first test to evaluate different machine learning and DL models in subsequent experiments.

To evaluate and compare the performance of the proposed models, the specificity (true negative rate), the sensitivity (true positive rate), and the area under the curve (AUC) were the main metrics used. For an additional comparison in some experiments, the GM and the equal error rate (EER) were also computed.

### 4.1. R10Fold and LOSO Evaluation 

Some authors have used random 10-fold cross-validation to analyze the performance of the proposed FOG detection systems. In this methodology, only the data of one patient is used in training and testing in order to make personalized detection models [44,59,64]. The overall results are calculated by averaging the results achieved from each patient.

Despite being a widely used methodology, the R10Fold evaluation could show overly optimistic performance when using overlapped windows, because there is a high probability that the training and test subsets share some of the same data, because of the random splitting process [67].

In order to improve the evaluation of R10fold evaluation with overlapped windows, a more precise evaluation can be made by using a LOSO cross-validation. In addition, this methodology allows the development of generic FOG detection systems which may be used in new patients without the need to acquire new data or retrain models. 

In LOSO evaluation methodology a training subset is created with data from all patients except one that will be used for testing. The entire process is repeated several times using a different patient for testing [59,64,67].

For this work, the LOSO evaluation was made for each of the 21 patients, then the entire process has been repeated six times to verify the variability. The reported results were computed by averaging all the experiments.

### 4.2. Baseline and Best Data Representation

To make a baseline, three data representations with three overlap percentages were analyzed by using a Random Forest classifier with 100 estimators. The experiments made in this subsection were tested by both R10fold and LOSO evaluations for additional comparison.

The first data representation is the one proposed by Mazilu [44], which has been considered as a standard baseline in several studies because of the easy implementation and excellent results. The second data representation tested was MFCCS adapted to low frequency [52]. The third data representation was a 64-bin FFT. 

#### 4.2.1. R10fold Evaluation with Random Forest (Personal Model)

Table 3 presents the results of sensitivity, specificity, and AUC with an R10fold evaluation for a personalized model. The sensitivity and specificity were calculated using an equal error threshold. The three-data representation present similar performances when using the same overlap. The best results were achieved by using Mazilu representation with a 75% overlap.

According to Table 3, sensitivity, specificity, and the AUC increase progressively as the percentage of overlap increases in all data representations tested. The results are expected because a greater amount of overlap can increase the probability of sharing data between the training and test subset in R10fold evaluation.

When Mazilu features were reproduced, the test achieved 0.897 in sensitivity and specificity (with the highest overlap setting), in contrast to the 0.995 and 0.999 reported with Daphnet with a Random Forest classifier. These differences can be justified because, in Daphnet, two patients did not show FOG events and their results showed a sensitivity of 1.0. Furthermore, the activities were performed under controlled conditions different than the normal ADL used in this study. 

Meanwhile, San-Segundo [67] reported an AUC of 0.951 with MFFCS, which is very similar to the 0.955 AUC reported in the present work, using the same Random forest classifier with 100 estimators and overlap setting of 75%.

For the spectral representation (FFT), the GM of 88.22 (sensitivity 90.5 and specificity 86) reported by Camps [65] is slightly superior to this work’s GM of 86.1 (86.4% sensitivity and 85.9% specificity) using 0% overlap. Although GM is similar in both studies, the difference can be attributed to the use of a different evaluation methodology, and the difference in the number of sensors.

#### 4.2.2. LOSO Evaluation with Random Forest (Generalized Model)

Table 4 presents the performance metrics evaluated with a LOSO evaluation. The best results for a generalized model were achieved by using a 64-bin FFT. Most results improve when using overlapped windows. An additional EER metric was included for comparison.

The results reported in [44] with LOSO evaluation, were 66.25% for sensitivity and 95.68% for specificity (GM 79.62), but this study’s reproduction achieved 82.4% sensitivity and 82.1% specificity (GM 82.2). While the GM between sensitivity and specificity is similar, the differences are expected because of the different window size (4 s), and the number of estimators in Random Forest. In regard to the MFCCS, this study has achieved similar AUC (0.913) to that of the AUC (0.901) reported in [67] using the same classifier and with an overlap of 75%.

According to Table 4, the results obtained with LOSO evaluation are lower than the results obtained through R10fold (Table 3). This decrease is expected because, in the LOSO evaluation, there are no shared data between the training and test subsets, which makes this evaluation more robust.

Mazilu and 64-bin FFT data representations present a better performance in the corresponding R10fold and LOSO evaluations, in both cases with an overlap of 75%. 

According to these results, the FFT data representation with overlap 75% will be used in the following sections to test several classification algorithms with LOSO evaluation in order to get the best generalized model.

For comparison purposes, the results achieved from Mazilu features in LOSO evaluation will be used as a baseline in the following sections. In this study’s reproduction, Mazilu representation achieved a sensitivity of 0.826, specificity 0.825 (GM 82.5%), and AUC of 0.904 with Random forest classifier and windows with an overlap of 75%.

Similar performances have been reported by other authors using LOSO evaluation when reproducing the same system: San-Segundo et al. [67] AUC 0.900 (sensitivity 0.934, specificity 0.0939), Rodríguez-Martín et al. [59] GM 83.65% (sensitivity 94.75, specificity 73.72). 

### 4.3. Evaluation of Different Classification Algorithms

In this experiment, different classification algorithms and detection methods were tested using the 64-bin FFT data representation. The results of the novelty detection methods are presented in Section 4.3.1, and the results from the supervised methods are presented in Section 4.3.2. All the experiments made in this subsection were evaluated through LOSO evaluation.

#### 4.3.1. Novelty Detection Models 

Two novelty detection models were tested: OC-SVM and an MLP auto-encoder. In these classification methods, only data labelled as normal movement is used in training. In prediction, it is assumed that patterns with abnormal movement (with significant statistical differences from normal patterns) are FOG events. For OC-SVM, this study’s best results were achieved with linear kernel, while in the MLP auto-encoder hypermeters described in Section 3.4.1 were used.

Table 5 shows the performance metrics obtained with a LOSO evaluation, by using 64-bin FFT with an overlap of 75%. Figure 5 shows the mean curves of sensitive and specificity obtained from six repetitions of the entire 21 subjects LOSO evaluation.

According to Table 5, the MLP denoiser auto encoder presents a better performance over the OC-SVM. Nevertheless, the performance of the novelty detection method is lower than the results achieved with the supervised methods shown in Table 4.

#### 4.3.2. Supervised Classification Models

For testing the supervised methods, the performance of five algorithms was evaluated. The classifiers used in this experiment include SVM (with a radial basis function kernel, gamma equal to 0.3 and penalty parameter equal to 1), AdaBoost with 100 estimators, Random Forest with 100 estimators, CONV-MLP and CONV-LSTM. The hyperparameters used in the DL models are described in Section 3.4.2 and Section 3.4.3 respectively.

The results achieved in the supervised methods are shown in Table 6. Figure 6 shows the mean curve of sensitive and specificity for additional comparison.

According to Table 6, the CONV-LSTM deep network presents the best performance compared to other supervised methods. However, the results obtained with random forest and CNN-MLP present similar AUC results.

In this experiment, the shallow models present lower performances than the deep learning classifiers. SVM classifier presents the lower results (AUC 0.876) using a radial basis function kernel.

### 4.4. Evaluation of Different Contextual Windows with the CNN-LSTM

In the following experiment, the inclusion of features from previous temporal windows was tested, considering that FOG events present temporal phases that appear at the beginning of the FOG event [82,83], and considering the results achieved in the previous experiment. The CONV-LSTM model was used to take advantage of the capability of this architecture to work with sequential data, by means of the time distributed and recurrent layers. 

In the CONV-LSTM model proposed in this work, previous spectral windows can be modeled as a time-steps of a sequence that feeds the input of the time distributed layer. The addition of data from previous windows has shown an improvement in FOG detection, without the need of increasing the window size and avoiding increasing latency [65,67].

In this experiment, the effect of including previous windows has been tested with LOSO evaluation by using 64-bin FFT with a 75% overlap. Table 7 shows the results obtained from the test of the CONV-LSTM model when using the current window data and until three previous windows. Figure 7 shows the mean curves of sensitive and specificity for additional comparison.

The best results in this study with LOSO evaluation were obtained with the CNN-LSTM model when using the current and three previous spectral windows. According to Table 7, the addition of previous spectral windows gradually improves the discriminative power of the deep network. The AUC reaches a maximum value of 0.939 while the equal error rate is reduced to 12.9%.

The improvement archived in FOG detection with the spectra representation and the CNN-LSTM model present an opportunity to enhance the clinical outcomes with the use of wearable devices for a long-term and accurate gait analysis in related applications [66,84,85].

Further experiments with this system could include the validation in the automatic identification of abnormal gait patterns or postures with the use of specific sensor locations with the aim of improving the accuracy in classification tasks, based in the analysis of inertial signals to be used in real daily-living environments [85,86].

### 4.5. Results of the CNN-LSTM with Three Previous Windows Per Patient

For comparison purposes, Table 8 shows the results obtained with the best model presented in this paper, the data representation used is 64-bin FFT with three previous spectral windows.

Additional metrics such as GM and EER have been included. The results presented are the mean values of six iterations per patient with LOSO evaluation, in a similar manner to that of previous experiments.

According to Table 8, the mean sensitivity and specificity is 0.871 by using an equal error threshold instead of maximizing the GM between sensitivity and specificity. In most patients, the AUC value reaches 0.915 except for the patients 2 and 20, while the EER remains below 15% except in patients 2, 4, and 20. The results show a high performance in FOG detection in the majority of the patients. The partial results of the patients 2 and 20 tend to reduce the global results and this may be explained by the fact that some patients present different freezing patterns when they are at their homes or in different scenarios. However, the average result presents a high AUC performance for a generic model by using LOSO evaluation.

## 5. Discussion and Conclusions

In this paper, different approaches for FOG detection in home environment with a single tri-axial were evaluated. The classification methods were based on supervised and self-supervised (novelty detection) by using both ML and DL algorithms. First, three data representations were tested to compare the results with the literature and identify an accurate data representation. Then, variations of the FFT data representation with some previous temporal windows were tested in subsequent experiments.

The evaluation of the classifiers was carried out with a LOSO evaluation, in order to train and evaluate a generic model for FOG detection. In this methodology, the data of one patient was held out from the rest of the patients, and then, the process was repeated (six times) for each of the patients.

Several experiments were conducted in this study. Most of the experiments were evaluated under a set of standard metrics such as sensitivity, specificity, GM, AUC, and EER. 

Results show that using spectral features from previous windows and a classifier based on CNN and LSTM layers it is possible to enhance the performance of the FOG detection system in terms of accuracy while maintaining a low-latency time. These improvements are required to provide a precise and long-term monitoring system an allow the development of a low-latency detection system necessary to be used as a cue-system to reduce the FOG events without interrupting patients’ daily live.

A comparison and discussion of the experiments and results are presented in the following subsections.

### 5.1. Comparison between R10fold and LOSO for Different Representations

The three data representations tested in the first experiment (Section 4.2) show that the best data representation for a personalized model (R10fold evaluation) was Mazilu [44], however, the MFCCS and FFT representations present similar performances for sensitivity and specificity.

For a generic model with LOSO evaluation, the best results were achieved with the 64-bin FFT data representation extracted from 128 samples window, while MFCCs features show a significant improvement for generic models over the Mazuli features when using overlapped windows. These results suggest that spectral representations can achieve high performances with generic models, while the use of hand-crafted FOG specific features (e.g., Mazilu features) show an improvement in the performance of personalized models.

The results achieved with the LOSO evaluation show a decrease in the performance over the R10Fold, however, the LOSO methodology shows more rigorous and accurate results when working with overlapped windows, avoiding data sharing between the train and test subsets.

LOSO methodology allows the development of generalized detection systems for using on new patients with data not included in the study. The addition of new data to train personalized models can be difficult, because of the inherent time and resource consumption to collect and label new data.

According to Table 3 and Table 4, the performance increases when using overlapped windows in the three data representations and the two evaluation methodologies. Progressive improvement is presented in the LOSO evaluation, while in R10fold a significant improvement is presented when using higher overlap values, because of the random data sharing between the train and test subsets.

Given these results future studies should consider including the LOSO methodology to conduct robust evaluations of systems designed for medical use. Especially if data processing includes the analysis of time-signals with overlapped windows.

### 5.2. Comparison between Novelty Detection and Supervised Methods

In Section 4.3, several data classification algorithms were tested with LOSO evaluation and FFT with a 75% overlap as data representation. In Table 5 two novelty detection methods were compared. An improvement in the sensitivity and specificity is achieved using MLP Denoiser Autoencoder over the OC-SVM. This behavior was analyzed in [87], where OC-SVM achieved a lower performance in high-dimensional datasets.

In regard to supervised methods, Table 6 shows that the CNN-LSTM deep network presents the best performance with 0.849 in sensitivity and specificity. Random forest and CNN-MLP present a slightly lower performance to that of CNN-LSTM, but they show similar AUC results (0.921 and 0.920 respectively). Notwithstanding, the Random forest presents an improvement in sensitivity and specificity, as well as a lower EER. AdaBoost with 100 estimators and SVM classifiers present the weakest performance in the experiments for supervised methods.

According to Table 5 and Table 6, the supervised methods present a better performance than novelty detection. When training the novelty detection models, the data corresponding to FOG events are ignored, leaving only data from normal movements. This data reduction affects performance, reducing the generalization capacity of the algorithm. Despite these results, the use of novelty detection presents a reliable method for FOG detection and abnormal gait detection with a low computational burden, without the need for labeled data [66].

In both supervised and novelty detection methods, the deep learning approaches present improvement over the shallow machine learning approaches. According to the results, the use of a higher amount of data in training can improve the generalization capacity of the network, increasing the performance in the classification of data that has not been used previously in training. 

### 5.3. Analysis of the Use of Contextual Windows with the CNN-LSTM

According to the results shown in Table 7, when using previous windows through the CNN-LSTM model, there is a progressive increase in the performance when adding more previous spectral windows. The results are improved with a reduction in the EER from 15.1% (with no previous windows) to 12.99% with tree previous windows, while the AUC is increased from 0.923 (without previous windows) to 0.939 with three previous windows.

In the best model proposed in this study, an AUC of 0.939 and an ERR of 12.9% were achieved. To the authors’ knowledge, these are the highest values achieved for AUC, sensitivity, and specificity when using only a triaxial accelerometer located in the waist through the LOSO evaluation (generic model) in this data set, which considers a real environment with ADL instead of laboratory conditions.

The results suggest that using temporal information (previous windows) modeled through LSTM layers shows a further improvement in FOG detection without increasing the length of the window. In the proposed model the CNN layers work as a feature extractor, while the LSTM increases the classification capabilities by modeling the temporal phases of the FOG events.

Related applications could benefit from applying this approach to classification problems, taking advantage of temporary dependencies in time-series in order to make robust classifiers and predictors.

### 5.4. Analysis of the Use of CNN-LSTM with Three Previous Windows Per Patient

With regard to the per patient analysis, the results in Table 8 show AUC values that exceed 0.915, except for the patients 2, 4, and 20. These patients show a different pattern, making it difficult to identify FOG events though LOSO. Despite decreased performance in these patients, only patient 1 presents a sensitivity and specificity bellow 0.804.

The different FOG patterns between subjects present a challenge for making a system that maintains high performances. Also, imbalanced data in FOG detection present an additional challenge to make accurate and reliable systems.

In this study, the use of spectral representations with previous windows shows an accurate trade between performance on several patients. The result shows that wrapped CNN layers are able to identify different FOG patterns with high accuracy, even with a reduced number of samples. This finding encourages the validation of this system in applications related to outlier detection.

### 5.5. Comparison of the Base Line and the CNN-LSTM

When comparing the results obtained through LOSO, from the baseline supported in Mazilu features and Random Forest with this study’s best model based in CNN-LSTM with 64 bin FFT and three previous windows, one may observe an improvement in AUC from 0.904 to 0.936 and a 4.8% decrease in EER (from 17.5% to 12.7%), apart from sensitivity and a specificity increase from 0.826 to 0.871 and 0.825 to 0871, respectively.

The improvement achieved with this method over the base line also brings the opportunity to develop effective tools for clinical applications that can help in decision-making in tasks related to the gait analysis in an ambulatory environment. A low-latency system presents the chance to develop wearable devices to be used in real-time applications for health management and rehabilitation assessment [85,86].

In this paper, different approaches have been evaluated for FOG detection using a body worn IMU and automatic classification techniques in the patients’ homes. The proposed CONV-LSTM architecture shows a suitable method for FOG detection, considering previous temporal windows that provide temporal information about the initial phases before FOG events.

Also, the results show that this method is capable of working with signals from a single sensor. Using low number of sensors should be considered to make a suitable system to be used in ambulatory environments and to maintain a low energy consumption because of digital processing without interfering in the performance.

In this study, the supervised methods have overperformed novelty detection. The decrease in performance is due to fact that the data corresponding to FOG events are not considered in training and it is assumed that FOG events occur (but may not necessarily) as a deviation from a normal pattern. Despite these results, the use of novelty detection presents a feasible method for the fast detection of abnormal gait patterns in related applications [66].

Spectral data representations (e.g., MFFCs and FFT) show a high performance in generalized models evaluated through LOSO, over the hand-made features. In this study, the use of FFT spectral representation shows high generalization-capability to detect spectral features related to gait patterns that can be beneficial to train generalized models. The outcomes of these methods could be used in conjunction with standard gait parameters to provide reliable indicators for PD management.

DL approaches present a better performance than shallow algorithms in all of the present study’s experiments. A higher amount of data obtained from a high overlap setting have benefited the DL approaches, this amount of data has allowed the models to have a better generalization capacity to classify data from unseen patients.

This study’s best CONV-LSTM model with three previous spectral windows presents a reduction of the EER of 4.8% when comparing to the baseline method (Mazilu features with a Random forest classifier). The use of previous windows shows a progressive improvement in FOG detection without increasing the latency in order to use the system as an external cue. However, the use of more than three previous windows can be impractical for real-time applications, to avoid the computational burden, also the improvement tends to be more reduced at each additional time step.

The comparison of the results with most of the related literature work is difficult, because of the diversity of approaches and validation methodologies. The best results reported in similar studies using a single sensor on a free-living environment with LOSO evaluation are 79.03% and 74.67% [59] and 84.49% on sensitivity and 85.83 on specificity [64].

Using LOSO evaluation, the CONV-LSTM model with three FFT previous windows achieved an AUC value of 0.939 and a sensitivity and specificity of 87.1%, the highest values of AUC in FOG detection using a single triaxial accelerometer, to the authors’ knowledge. This approach can be used in a real situation (patients’ homes) when performing ADL, where it is expected an increase in the occurrence of FOG events.

Despite the high performance of the DL approaches in FOG detection, it is important to consider that training deep models with recurrent layers requires a large number of computational resources. Moreover, the best results with DL are achieved only when using a great amount of data (accurately labelled) which is a time-consuming process. To overcome these issues, it is possible to use data augmentation techniques or transfer learning, for example, using a dataset designed for human activity recognition [88].

Future research could also include the validation of the proposed DL models in a data base with a larger number of patients that show different FOG patterns. Further improvement in the performance could be made with the development of more complex networks and novel data representations based on time-frequency analysis.

The efficient training of DL models is an important issue to address. Novel loss-functions designed to work with imbalanced data like FOG or abnormal gait detection should be developed and validated to improve the model performance and optimize the training time.

Also, the validation of efficient algorithms and pre-processing techniques is an important research area to develop low-latency systems that can be used for early prediction, long-term monitoring, or for the development of real-time FOG detection devices that can help maintaining a certain gait speed with the use of external cues.

## Figures and Tables

**Figure 1 sensors-20-01895-f001:**
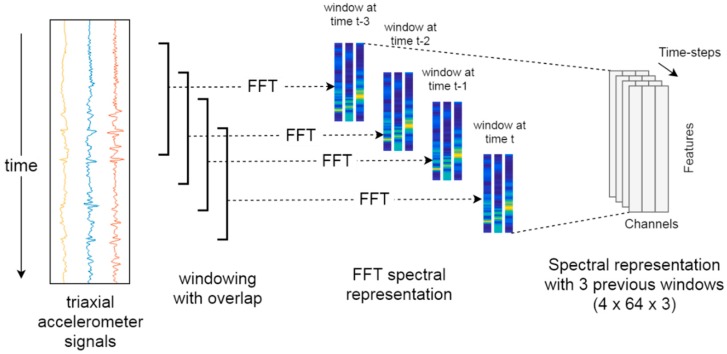
Fast Fourier transform (FFT) spectral representation with three previous windows.

**Figure 2 sensors-20-01895-f002:**
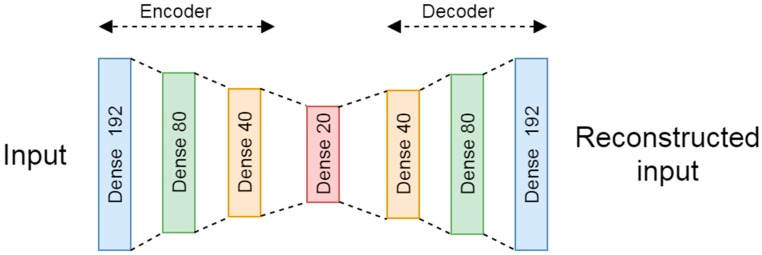
Denoiser auto encoder.

**Figure 3 sensors-20-01895-f003:**
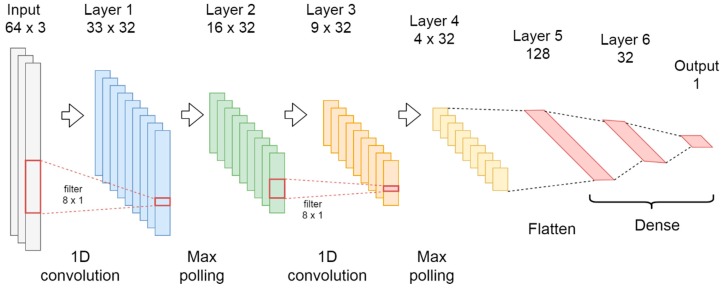
Deep neural network with convolutional layers (CNN).

**Figure 4 sensors-20-01895-f004:**
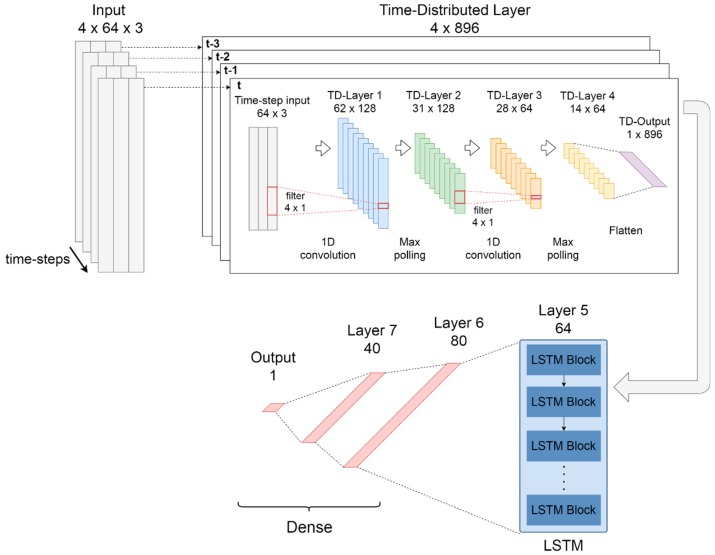
Deep neural networks with convolutional and long short-term memory (LSTM) layers (CNN-LSTM).

**Figure 5 sensors-20-01895-f005:**
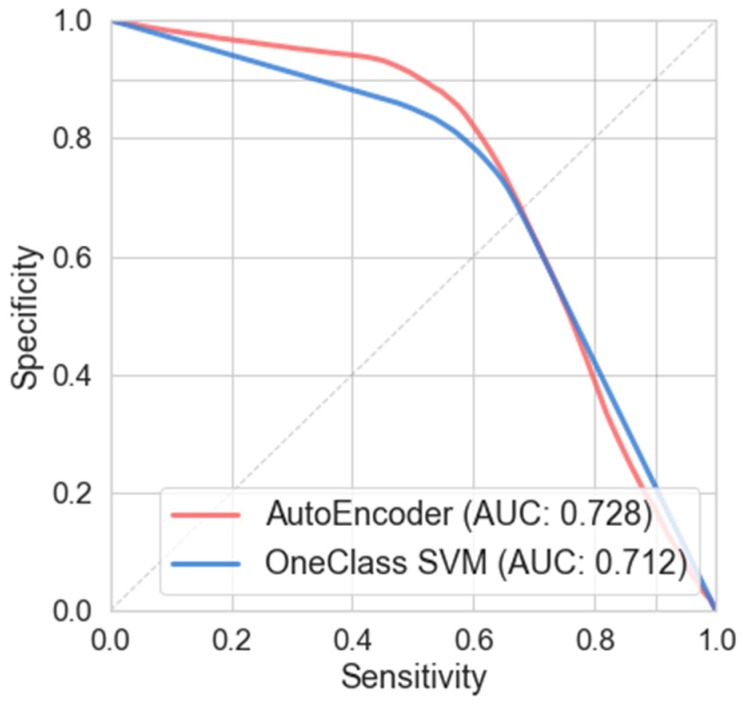
Specificity vs. sensitivity curves, an area under the curve (AUC), for two novelty detection algorithms with LOSO evaluation, using a 64-bin FFT data representation and 75% overlap.

**Figure 6 sensors-20-01895-f006:**
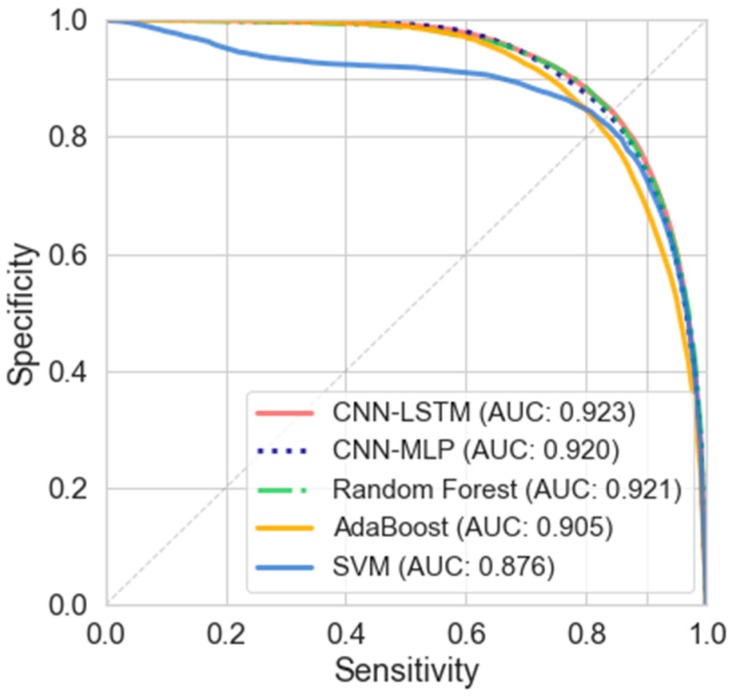
Specificity vs. sensitivity curves, and AUC for the supervised classification models with LOSO evaluation, FFT data representation with a 75% overlap.

**Figure 7 sensors-20-01895-f007:**
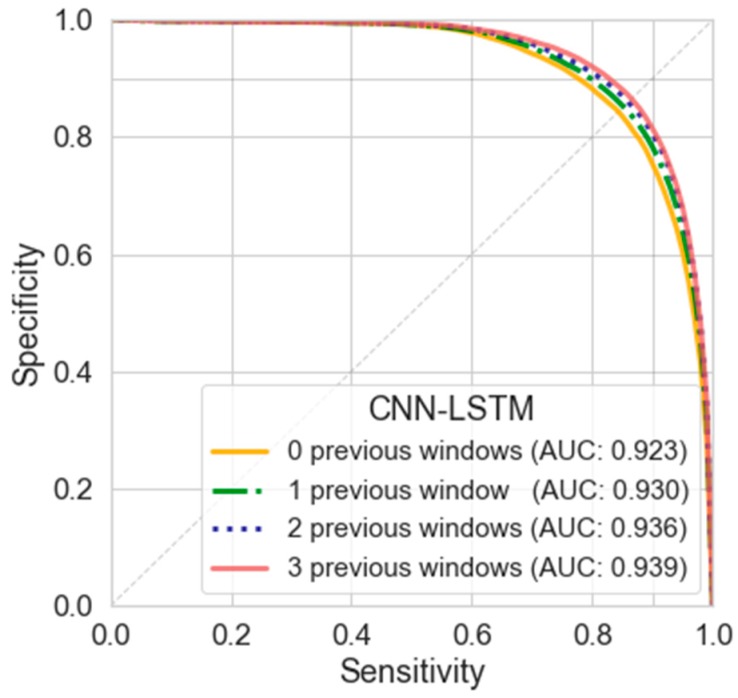
Specificity vs. sensitivity curves, and area under the curve (AUC) when including previous windows in the CNN-LSTM model, using a LOSO evaluation with FFT data representation and 75% overlap.

**Table 1 sensors-20-01895-t001:** Summary of highlighted works regarding freezing of gait (FOG) detection with wearable sensors.

Publication	Analysis Methods	Sensors and Location	Participants	Main Results
Moore et al. [14]	Threshold method to analyze the power of specific frequency bands	Accelerometers located in the shank	11 PD	Identification of the increasing of power in specific frequency bands when FOG appears. Detection of 78% of FOG events.
Bächlin et al. [60]	Threshold analysis from three different frequency bands	9 accelerometer signals from Daphnet [68]. sensors located in the ankle, knee, and lower back	10 PD (8 with FOG)	Reduction of false detections with the addition of a Total-band threshold. A sensitivity of 73.1% and specificity of 81.6%.
Mazilu et al. [44]	ML techniques with Bächlin et al. [60] and statistical features	9 accelerometer signals from Daphnet [68]	10 PD (8 with FOG)	Sensitivity and specificity over 95% with 10fold cross-validation. A sensitivity of 66.25% and specificity of 95.38% with LOSO cross-validation.
Moore [61]	Threshold analysis	7 sensors located at the lumbar back, thighs, shanks, and feet	25 PD	Identification of the shank and back as the most convenient places to the sensors. Sensitivity 84.3% specificity 78.4%.
Tripoliti et al. [62].	ML techniques in a four steps method	6 accelerometers and 2 gyroscopes attached to different parts of the body	16 People (5 healthy, 6 PD with no FOG, and 5 with FOG)	Sensitivity of 89.3% and specificity of 79.15% with LOSO evaluation considering only patients with FOG symptoms.
Zach et al. [63]	Threshold detection with Moore et al. [14] features	A single triaxial accelerometer placed at the waist	23 PD patients with FOG	A lumbar sensor is identified as the best place for FOG detection. A sensitivity of 75% and specificity of 76%.
Ahlrichs et al. [15]	SVM classifier with frequency and statistical features	Single waist-worn sensor with a triaxial accelerometer	20 PD (8 with FOG and 12 with no FOG)	Frequency-based features could be reliably used to detect FOG. A sensitivity of 0.923, and specificity of 1 using data from 5 patients for testing.
Rodríguez-Martín et al. [59]	SVM classifier with statistical and spectral features validated with R-10fold and LOSO	Single waist-worn sensor with a triaxial accelerometer	21 PD	A sensitivity 88.09% and specificity 80.09% with R-10fold cross-validation, and a sensitivity of 79.03% and specificity of 74.67% for LOSO evaluation.
Samà et al. [64]	ML algorithms with a reduced version of the features proposed by Rodríguez-Martín et al. [59]	Single waist-worn sensor with a triaxial accelerometer	15 PD	Systematical reduction of the number of features. A sensitivity of 91.81% and specificity 87.45% for R-10-fold, and sensitivity of 84.49% and specificity 85.83% in LOSO evaluation.
Camps et al. [65]	DL and ML techniques. A novel spectral data representation	9-channel waist-worn IMU with accelerometer, gyroscope, and magnetometer	21 PD	The use of CNN with novel spectral data representation. AUC of 0.88, a sensitivity of 91.9 and a sensibility of 89.5 when testing with data of 4 patients.
Mohammadian et al. [66]	Novelty detection with CNN denoising autoencoders	9 accelerometer signals from Daphnet [68]	10 PD (8 with FOG)	Validation of a method to detect abnormal movement without the need for labeled data for training. Average AUC of 0.77.
San-Segundo et al. [67]	DL and ML algorithms validated in four different data representations	9 accelerometer signals from Daphnet [68]	10 PD (8 with FOG)	Validation of DL-based systems with CNN with a novel MFCC data representation. The analysis of the use of previous and posterior windows. AUC of 0.931 and an EER of 12.5% with LOSO cross-validation.

**Table 2 sensors-20-01895-t002:** Summary of the data representations used in the current study.

Data Representation	Number of Features per Signal	Description of the Features
Mazilu [44]	7	Mean, standard deviation, variance, frequency, entropy, energy, freeze index (power of the freezing band 3–8 Hz divided by power in locomotor band 0.5–3 Hz), and the sum of freeze index and locomotion band.
MFCCS [52]	12	Mel frequency cepstral coefficients adapted to inertial signals.
FFT [53]	64	Symmetric part of FFT N = 128
FFT + one previous window	128	Symmetric part of FFT N = 128, plus 1 previous spectral window.
FFT + two previous windows	192	3 × Symmetric part of FFT N = 128, plus 2 previous spectral windows.
FFT + three previous windows	256	4 × Symmetric part of FFT N = 128, plus 3 previous spectral windows.

**Table 3 sensors-20-01895-t003:** R10Fold evaluation (personalized model) with a Random forest classifier with 100 estimators for different data representations and windows overlapping percentages.

Data Representation	Overlap	Sensitivity	Specificity	AUC
Mazilu	0%	0.865	0.866	0.932
	50%	0.883	0.881	0.944
	75%	0.897	0.897	0.957
MFCCS	0%	0.867	0.856	0.922
	50%	0.878	0.876	0.940
	75%	0.894	0.894	0.955
FFT	0%	0.864	0.859	0.929
	50%	0.877	0.877	0.943
	75%	0.894	0.894	0.956

**Table 4 sensors-20-01895-t004:** Leave-one-subject-out (LOSO) evaluation with a Random forest classifier with 100 estimators for different data representation and window overlapping.

Data Representation	Overlap	Sensitivity	Specificity	AUC	EER (%)
Mazilu	0%	0.824	0.821	0.900	18.0
	50%	0.829	0.826	0.903	17.4
	75%	0.826	0.825	0.904	17.5
MFCCS	0%	0.824	0.824	0.909	17.6
	50%	0.831	0.827	0.911	17.3
	75%	0.835	0.834	0.913	16.7
FFT	0%	0.842	0.839	0.916	16.1
	50%	0.841	0.839	0.919	16.1
	75%	0.849	0.847	0.921	15.3

**Table 5 sensors-20-01895-t005:** LOSO evaluation results of the novelty detection models. 64-bin FFT and overlap 75%.

Classifier	Sensitivity	Specificity	GM	AUC	EER (%)
OneClass-SVM	0.832	0.592	0.702	0.712	40.8
MLP Autoencoder	0.694	0.694	0.694	0.728	30.6

**Table 6 sensors-20-01895-t006:** LOSO evaluation results of the supervised classification models with a data representation of 64-bin FFT and overlap 75%.

Classifier	Sensitivity	Specificity	GM	AUC	EER (%)
SVM	0.832	0.831	0.832	0.876	16.9
AdaBoost	0.827	0.828	0.827	0.905	17.2
Random Forest	0.849	0.847	0.848	0.921	15.3
CNN-MPL	0.844	0.844	0.844	0.920	15.6
CNN-LSTM	0.849	0.849	0.849	0.923	15.1

**Table 7 sensors-20-01895-t007:** Results of the CNN-LSTM model with a different number of contextual windows.

Previous FFT Windows	Sensitivity	Specificity	GM	AUC	EER (%)
0	0.849	0.849	0.849	0.923	15.1
1	0.858	0.858	0.858	0.930	14.2
2	0.867	0.867	0.867	0.936	13.3
3	0.871	0.871	0.871	0.939	12.9

**Table 8 sensors-20-01895-t008:** Results per patient of the CNN-LSTM with three previous windows using LOSO evaluation.

Patient Index	Sensitivity	Specificity	GM	AUC	EER (%)
1	0.911	0.910	0.910	0.970	8.957
2	0.683	0.683	0.683	0.767	31.745
3	0.912	0.911	0.911	0.959	8.890
4	0.808	0.807	0.808	0.904	19.307
5	0.849	0.848	0.849	0.925	15.153
6	0.881	0.880	0.881	0.946	11.960
7	0.880	0.877	0.879	0.946	12.270
8	0.879	0.878	0.878	0.925	12.175
9	0.878	0.879	0.879	0.954	12.080
10	0.892	0.893	0.892	0.956	10.741
11	0.836	0.836	0.836	0.916	16.406
12	0.867	0.868	0.868	0.940	13.201
13	0.862	0.862	0.862	0.948	13.803
14	0.928	0.927	0.928	0.980	7.289
15	0.942	0.942	0.942	0.981	5.807
16	0.885	0.886	0.885	0.945	11.440
17	0.884	0.884	0.884	0.952	11.631
18	0.916	0.916	0.916	0.973	8.370
19	0.896	0.897	0.897	0.965	10.320
20	0.804	0.804	0.804	0.909	19.560
21	0.894	0.895	0.895	0.958	10.459
Average	0.871	0.871	0.871	0.939	12.932

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
