# Peer review of "Deep Learning Approaches for Detecting Freezing of Gait in Parkinson’s Disease Patients through On-Body Acceleration Sensors"

_sensors, 2020, doi:10.3390/s20071895_

Round 1

Reviewer 1 Report

Sigcha et al. applied machine and deep learning for the detection of freezing of gait in patients with Parkinson's disease wearing inertial measurement unit at the waist. Currently, FOG is evaluated via questionaires and during a brief visit at the doctor's office. Objective and reliable measurements are lacking. Constantly monitoring gait in a home environment and subsequent analysis could close this gap. A high AUC in FOG detection was achieved using the CONV-LSTM model. 

Overall, the research question is of high interest, and comparison of different deep learning methods. 

There are just a few points the authors need to address. The introduction needs to be shortened, information not related to the topic should be removed, e.g. sentences about surgical procedures and Parkinson. The background section needs to be more concise, a table highlighting previous works would be beneficial.

LOSO evaluation needs to be introduced and explained.

How long did the patients wear the IMU? Was this performed at home? What were the results of the questionnaires? 

Author Response

All the issues asked by the reviewer are listed point by point in the attached document.

Kind regards.

Reviewer 2 Report

Given the increasingly more widespread adoption of wearable sensors due to their autonomous, low cost, easy to operate, I find it interesting to explore the applicability of wearable device for gait detection. In this study, different approaches based on machine and deep learning for FOG detection in patients' homes are evaluated, resorting to a leave-one-subject-out (LOSO) cross-validation, using only acceleration signals from a body-worn Inertial Measurement Unit (IMU). The experiment results show that the current method has the potential for clinical gait analysis in Parkinson's Disease Patients. And the authors claimed that this research potentially opened the way to novel research studies, and enhancing the performance of FOG detection systems to be used in real home-environments can help to optimize the PD management and treatments (pharmacologic and non-pharmacologic.

Generally, the paper is well written and quite interesting to my mind, however, the authors have to make their scientific or technological contributions clear. It is absolutely not enough to have a manuscript accepted by just reporting what they have done. Thus the authors should make their contributions logical and clearer. For further details, please refer to the following statements:

1. The abstract is a bit poor. It is not enough to just list the methods that were employed in the manuscript. Furthermore, in line 23, “The results show an improvement in FOG detection when using deep-learning methods over shallow machine learning” is not a good way to explicitly state the main points of this paper. The authors are suggested to make the problems and their contributions outstanding.

2. In the Materials and Methods part, there is a lack of information regarding the validation experiment such as the detailed of subjects involved.

3. I noticed that in line 408 “the best performance in the training process was 150 epochs, batch size equal to 512”, the authors should explain how these parameters were obtained and what would happen if this configuration was set improperly.

4. The authors have to clearly show how this proposal is advancing knowledge in human gait monitoring field and its applications in clinical practice, which is mentioned in the abstract but not explained explicitly in the Experiments section from the perspective of medical applications.

5. More comparison with the literature may be provided in the experimental results section. Some examples are:
- Anwary, A. R., Yu, H., & Vassallo, M.. Optimal Foot Location for Placing Wearable IMU Sensors and Automatic Feature Extraction for Gait Analysis. IEEE Sensors Journal, 2018, 18(6), 2555–2567.
- Qiu, S., Wang, H., Li, J., Zhao, H., Wang, Z., Wang, J., … Ru, B.. Towards Wearable-Inertial-Sensor-Based Gait Posture Evaluation for Subjects with Unbalanced Gaits. Sensors (Basel, Switzerland), 2020, 20(4), 1–18.
- Wang, Z., Li, J., Wang, J., Zhao, H., Qiu, S., Yang, N., & Shi, X.. Inertial Sensor-Based Analysis of Equestrian Sports Between Beginner and Professional Riders Under Different Horse Gaits. IEEE Transactions on Instrumentation and Measurement, 2018, 67(11), 2692–2704.

6. What are the implications of the findings? More discussion should be provided in the manuscript. It is not enough to have a paper accepted by just reporting what they have done.
Hence their contribution to the field is still unclear, and it requires substantial improvement before publication, thus the authors should make their contributions more clearly.

7. There are a few typos and minor English errors scattered throughout the paper. Please proofread the manuscript to improve the readability.
Just name a few, in line 14, “Freezing of gate (FOG) is one of the most incapacitating motor symptoms”, where “gate” should be “gait”.
In line 401, “The first convolutional layer has 32 filters and a kernel size of 8, The second layer…”
Capital letter was presented behind the comma.

Based on the above reasons, I acknowledge the merits of this article but would recommend a revision before publication.

Author Response

(The authors gave the same response as above.)

Reviewer 3 Report

Lines 33-57 of the Introduction contain statements that are either incorrect, sophomoric or unnecessary and should be deleted.

It would be helpful if the authors summarized the kinematic features of FOG that are being extracted and used in the Daphnet dataset.

The last paragraph states “In future work, improvement in FOG detection can be done by increasing the amount of data, by adding more patients and more time besides the data augmentation. Future research could also include the use of generative adversarial networks and novel data representations to improve the energy saving of the devices, in order to make accurate systems to be used as cue devices or for long term monitoring.” I don’t understand most of this, and most importantly, I am skeptical of the statement in the first sentence.

Author Response

(The authors gave the same response as above.)

Round 2

Reviewer 2 Report

I thank the authors for their modifications, clarifications, and their continued work on improving this manuscript. While I value the authors efforts to answer all my concerns and feel that the paper overall improved. All the previous concerns were well addressed in the revised manuscript. Overall, the revised paper quality meets the requirements of Journal of "Sensors", hence the manuscript could be accepted for publication.